

# Evolution of the ESA CCI Soil Moisture Climate Data Records and their underlying merging methodology

Alexander Gruber[1,2], Tracy Scanlon[2], Robin van der Schalie[3], Wolfgang Wagner[2], and Wouter Dorigo[2]

[1]Department of Earth and Environmental Sciences, KU Leuven, Heverlee, Belgium
[2]Department of Geodesy and Geoinformation, TU Wien, Vienna, Austria
[3]VanderSat, Haarlem, The Netherlands

**Correspondence:** A. Gruber (alexander.gruber@kuleuven.be)

**Abstract.** The European Space Agency's Climate Change Initiative for Soil Moisture (ESA CCI SM) merging algorithm generates consistent quality-controlled long-term (1978 - 2018) Climate Data Records for soil moisture which serves thousands of scientists and data users worldwide. It harmonises and merges soil moisture retrievals from multiple satellites into (i) an active-microwave-based only, (ii) a passive-microwave-based only, and a combined active-passive product, which are sampled
to daily global images on a 0.25 degree regular grid. Since its first release in 2012 the algorithm has undergone substantial improvements which have so far not been thoroughly reported in the scientific literature. This paper fills this gap by reviewing and discussing the science behind the three major ESA CCI SM merging algorithm versions 2 (http://dx.doi.org/10.5285/3729b3fbbb434930bf65d82f9b00111c; Wagner et al., 2018), 3 (http://dx.doi.org/10.5285/b810601740bd4848b0d7965e6d83d26c; Dorigo et al., 2018a), and 4 (http://dx.doi.org/10.5285/3a8a94c3fa464d68b6d70df291afd457; Dorigo et al., 2018b) and pro-
vides an outlook to the expected improvements planned for the next algorithm version 5.

## 1   Introduction

The European Space Agency's Climate Change Initiative for Soil Moisture (http://www.esa-soilmoisture-cci.org/), hereafter referred to as ESA CCI SM, is dedicated to the development of consistent satellite-based long-term Climate Data Records
(CDRs) for soil moisture, aiming to serve climate science as well as numerous other communities (Dorigo et al., 2017). The first soil moisture CDRs produced by the ESA CCI SM were released in 2012. To date the ESA CCI SM serves more than 5000 registered users, providing the basis for a host of scientific publications and data set applications (Dorigo et al., 2017).

Core to the ESA CCI SM is a merging algorithm which, in essence, merges soil moisture retrievals from various satellites with finite lifetimes and significantly varying instrument characteristics (frequency, spatial resolution, temporal coverage, po-
larisation, revisit time, etc.) into three consistent multi-decadal data sets. This process faces innumerable scientific challenges and is therefore subject to continuous research and development. To date, seven product versions have been released to the general public. Differences between these product versions range from minor bug fixes and data set extensions to major structural



changes in the way different satellite products are harmonised and merged. Recently, the European Commission's Copernicus Climate Change Service (C3S) started operational near-real-time CDR production, based on the algorithm developed within the ESA CCI SM.

While product improvements have been validated in numerous publications and the data set has been proven to be useful in
a large number of applications (for a comprehensive review of these studies see Dorigo et al. (2015, 2017)), none of the major scientific advances of the merging algorithm (since its first release in 2012; Liu et al., 2011, 2012; Wagner et al., 2012) have as yet been thoroughly documented in the scientific literature. This paper aims to fill this gap by providing a comprehensive and complete resource of the evolution of the ESA CCI SM merging algorithm up to the current version 4.4 which was released at the end of 2018. Moreover, an outlook to the expected developments that are planned for the next iteration, version 5, is
provided.

## 2   Evolution of the ESA CCI SM merging algorithm

The ESA CCI SM merging algorithm produces three individual products: (i) the ACTIVE product, which is generated by merging soil moisture retrievals from active-microwave-based sensors only, (ii) the PASSIVE product, which is generated by merging soil moisture retrievals from passive-microwave-based sensors only, and (iii) the COMBINED product, which is
generated by merging soil moisture retrievals from both active-microwave-based and passive-microwave-based sensors. This paper reviews the three major ESA CCI SM merging algorithms which have been utilised in released versions to date:

– The initial merging algorithm proposed by Liu et al. (2011, 2012) which has been used to generate all products up until version v02.2 (released early 2016; Wagner et al., 2018), henceforth referred to as ESA CCI SM v2. This algorithm is a decision tree based approach that selects either passive-microwave-based soil moisture retrievals, active-microwave-
based soil moisture retrievals, or computes an arithmetic mean of the two based on their mutual correlation and vegetation optical depth.

– The algorithm that has been used to generate the product versions v03.2 and v03.3 (released at the beginning and the end of 2017, respectively; Dorigo et al., 2018a) and for generating all Copernicus C3S CDR products up to version v201801, henceforth referred to as ESA CCI SM v3. This algorithm is based on a statistically rigorous Least Squares merging
approach.

– The algorithm that has been used to generate product versions v04.2 and v04.4 (released at the beginning and the end of 2018, respectively; Dorigo et al., 2018b) and is being used in the current Copernicus C3S CDR production system version (v201812), henceforth referred to as ESA CCI SM v4. This algorithm uses an improved uncertainty characterisation approach to better parametrise the Least Squares merging scheme.

Note that the official version numbering system (vX.X) of the ESA CCI SM products follows the convention that the first (two-)digit number denotes the version of the underlying data merging methodology while the second number marks releases with simple bug fixes and data set extensions.



Section 3 describes the Level-2 (L2) soil moisture retrieval algorithms and the pre-processing steps of the individual input data sets, which are generally common to all ESA CCI SM product versions. Section 4 provides a review of the initial merging algorithm proposed by Liu et al. (2011, 2012) and was used in ESA CCI SM v2. Section 5 discusses the limitations of this decision tree based algorithm and the theoretical requirements on a statistically optimal merging scheme. Section 6 describes
how such a statistical (Least Squares) merging scheme was implemented in ESA CCI SM v3 and Section 7 describes the improved uncertainty characterisation for this scheme that has been employed in ESA CCI SM v4. Section 8 demonstrates the performance evolution of the algorithm versions through comparison against ground reference data. Finally, Section 9 concludes with a discussion on the limitations and known issues with the current merging algorithm, which are currently under investigation and expected to contribute to the next released version of the dataset.

## 3   Input data and preprocessing

All ESA CCI SM algorithms to date merge pre-processed L2 data, that is, gridded soil moisture products retrieved from radiometrically calibrated backscatter or brightness temperature measurements. These data are resampled to a 0.25 degree regular grid using a hamming-window approach and to daily time stamps (00:00 UTC) using a nearest neighbour search. Currently (at ESA CCI SM v4), soil moisture products from four active-microwave-based instruments (in short: active products)
and seven soil moisture products from passive-microwave-based instruments (in short: passive products) are merged into the ESA CCI SM data sets. Sensors/missions, their temporal availability, and (most relevant) characteristics are summarised in Table 1.

### 3.1   Active products

Active products are retrieved using the TU Wien Water Retrieval Package (WARP) algorithm (Wagner et al., 1999; Naeimi
et al., 2009) which is also used to generate the official operational ASCAT L2 soil moisture products for the Satellite Application Facility on Support to Operational Hydrology and Water Management of the European Organisation for the Exploitation of Meteorological Satellites (EUMETSAT H SAF, http://hsaf.meteoam.it/soil-moisture.php). The WARP algorithm is a change detection approach that retrieves soil moisture as degree of saturation by scaling azimuthally corrected radar backscatter measurements between the historically lowest and highest observed values (at each individual grid location) which are assumed
to represent completely dry and saturated conditions, respectively. The multi-antenna multi-incidence angle capability of the ERS and ASCAT scatterometers are exploited to correct for the (seasonally varying) backscatter contribution of vegetation (Vreugdenhil et al., 2016). A threshold-based decision tree algorithm is applied to incidence angle normalized backscatter measurements to detect and remove measurements that were taken under frozen or freezing/thawing conditions where no reliable soil moisture retrieval is possible (Naeimi et al., 2012). For a complete description of the model and how it is applied
to ERS and ASCAT data see the "Algorithm Theoretical Baseline Document (ATBD) D2.1 Version 04.4 - Soil Moisture Retrievals from Active Microwave Sensors" (Chung et al., 2018b). Information as to which WARP algorithm versions have been used for the different ESA CCI SM versions can be found in Table 1.



## 3.2 Passive products

Passive products are retrieved using the Land Parameter Retrieval Model (LPRM) algorithm (Owe et al., 2008). LPRM is a forward model based on the radiative transfer model of Mo et al. (1982) and distinguishes itself from other soil moisture retrieval algorithms due to its applicability to a wide range of frequencies (i.e. 1-20 GHz) and by using an analytical solution based on the Microwave Polarization Difference Index for the derivation of the Vegetation Optical Depth (VOD; Meesters et al., 2005). For C-band and higher frequency sensors, the data is filtered for frozen conditions using Ka-band based temperature (Holmes et al., 2009) and for Radio Frequency Interference (RFI; de Nijs et al., 2015; Li et al., 2004). For SMOS L-band retrievals, the filtering is based on the RFI and modelled (by the European Centre for Medium Range Weather Forecasts) temperature data provided alongside with the brightness temperature input data (SMOS L3TB). For a complete description of LPRM and how it is applied see the "Algorithm Theoretical Baseline Document (ATBD) D2.1 Version 04.4 - Soil Moisture Retrievals from Passive Microwave Sensors" (de Jeu et al., 2018) as well as van der Schalie et al. (2016) and van der Schalie et al. (2017). Information as to which LPRM algorithm versions have been used for the different ESA CCI SM versions can be found in Table 1.

## 3.3 GLDAS-Noah

GLDAS-Noah (Rodell et al., 2004) is used as scaling reference in the COMBINED product to obtain a consistent climatology throughout the entire ESA CCI SM period (Liu et al., 2011, 2012, see the next sections) and as an instrumental product for triple collocation analysis in the ACTIVE, the PASSIVE and the COMBINED products to obtain the L2 input data uncertainties (Su et al., 2014). More specifically, GLDAS-Noah version 2.1 (and previously, GLDAS-Noah version 1), which provides data from 2000 until the present, is used for both rescaling and triple collocation analysis (see the next sections). In earlier periods, GLDAS-Noah version 2.0, which provides data from 1948 until 2010, is used for triple collocation analyses for L2 products where there is no temporal overlap with GLDAS-Noah v2.1 (or v1). However, all L2 data sets are rescaled (for the COMBINED product) against GLDAS-Noah v2.1 (previously, v1) due to inconsistencies with GLDAS-Noah v2.0, which originate from the historic forcing data that were used in this version.

GLDAS-Noah (all versions) provides 3-hourly estimates of soil moisture and other land variables for four different depth layers on a 0.25 degree regular grid. Top layer (0-10 cm) soil moisture estimates at 00:00 UTC (coinciding with the resampled satellite input data) are used for rescaling and triple collocation analysis. Top layer soil temperature ($T_s$) and snow water equivalent (SWE) estimates are used to mask out satellite measurements that were taken under conditions where no reliable soil moisture retrieval is possible (i.e, $T_s < 0°$ and SWE $> 0$ mm).





## 4 ESA CCI SM v2

As previously mentioned, the merging algorithm for all product versions up until v2 (released early 2016; Wagner et al., 2018) remained relatively unchanged and is described in detail in Liu et al. (2011, 2012). Therefore, only its key features will be summarised here. The principal steps of the algorithm are illustrated in Figure 1 and described in the following sections.

### 4.1 Harmonisation of L2 data

L2 input data sets from different missions are harmonised to a common climatology by matching the Cumulative Distribution Function (CDF) of the individual data sets to that of a reference product, which was chosen to be the one that is expected to have the most stable climatology. For the active products this is ASCAT as it is the direct successor instrument of ERS, with improved spatial resolution, temporal coverage and radiometric accuracy (Naeimi et al., 2009). For passive products it is AMSR-E due to its longer signal wavelength and higher spatial and temporal resolution (Liu et al., 2012). The CDF-matching is realized by splitting the data sets into percentiles and rescaling these percentiles in a linear fashion (Liu et al., 2011).

For the harmonisation of the active products, a combined ERS-1/2 50 km (native) resolution time series is first simultaneously retrieved from well inter-calibrated backscatter measurements from ERS-1 and ERS-2. Data gaps in these time series due to the on-board storage failure of ERS-2 in 2003 are filled with the experimental higher-resolution 25 km ERS-2-only soil moisture retrievals after rescaling (CDF-matching) them against the combined ERS-1/2 data set. This complete ERS time series is then rescaled against ASCAT.

Passive products are harmonised as follows. WindSat and TMI retrievals are rescaled against AMSR-E while AMSR2 is assumed to be properly inter-calibrated with AMSR-E (which is most likely not always the case; see Parinussa et al. (2015) and Section 9). For SSM/I, only anomalies (calculated as deviations from the mean seasonal cycle) are rescaled against those of AMSR-E while the mean seasonal cycle of SSM/I is fully replaced with that of AMSR-E due to its lacking consistency with other sensors (Liu et al., 2011, 2012). A merged SSM/I - TMI - AMSR-E time series is then created by selecting the best available sensor at a given time, assuming that the data quality negatively correlates with microwave frequency and the time since launch (Liu et al., 2012). Since TMI data are only available between $\pm 37$ degree latitude, SSM/I retrievals are used in the remaining latitude bands even though they are considered to be less reliable than those from the longer-wavelength TMI sensor. Finally, SMMR observations are rescaled against the merged SSM/I - TMI - AMSR-E time series. Note that, as there is no temporal overlap between SMMR and successive sensors, this step scales the observations of different periods and thus assumes that there is no trend from the SMMR period (1978 - 1987) to the combined TMI - SSM/I - AMSR-E period (1987 - 2011) (Liu et al., 2012).

### 4.2 Generation of the ACTIVE and the PASSIVE product

The ESA CCI SM v2 ACTIVE product is generated by concatenating ASCAT and the rescaled combined ERS time series. Since this product is provided in the ASCAT data space, i.e. as degree of saturation, a porosity map derived from the Har-



monized World Soil Database (HWSD; Nachtergaele and Batjes, 2012) is provided alongside the data in order to allow for converting the soil moisture estimates to volumetric units as required.

The ESA CCI SM v2 PASSIVE product is generated by concatenating the rescaled SMMR data set, the harmonised and already concatenated AMSR-E - TMI - SSM/I time series, and AMSR2. Note that in the ESA CCI SM v2 ACTIVE and
PASSIVE products measurements from different sensors are not merged. At all time steps, the presumed best-performing sensor operational at that date is selected, disregarding potentially available observations from the other sensors. Figure 2 illustrates the resulting sensor selection per time and latitude.

## 4.3 Harmonisation of the ACTIVE and the PASSIVE product

The ACTIVE and the PASSIVE products are harmonised by rescaling them against GLDAS-Noah soil moisture simulations.
The rational for using a land surface model as scaling reference for harmonisation is its supposedly long-term consistent climatology. For more details on the implications and caveats of this choice see Liu et al. (2011, 2012) and Section 9.

## 4.4 Generation of the COMBINED product

The harmonised ACTIVE and PASSIVE data sets are merged into the COMBINED product by following a decision tree that selects either one of the products alone or uses the arithmetic mean of both during a particular period based on the assumption
that active retrievals tend to perform better in more densely vegetated areas whereas passive retrievals tend to perform better in more sparsely vegetated areas (Liu et al., 2011, 2012). For each merging period (see Figure 2) and at each quarter degree grid cell, the Pearson correlation coefficient between the ACTIVE and the PASSIVE product is calculated. At all locations and during each period where the correlation exceeds 0.65, the arithmetic mean between the ACTIVE and the PASSIVE product is used at time steps where both are available. If one data set does not provide a valid observation at a particular time step
(due to L2 quality control and orbit characteristics), the observation of the respective other data set is used. At locations and in periods where the correlation threshold is not met, the ACTIVE data set is selected if the multi-year average AMSR-E based Vegetation Optical Depth (VOD) at that location is above a certain threshold and the PASSIVE data set is selected if the VOD is below that threshold. The threshold is taken as the average of multi-year average AMSR-E based VOD estimates over all regions where the correlation between the ACTIVE and the PASSIVE products does exceed the aforementioned correlation
threshold of 0.65 (i.e. where the ACTIVE and the PASSIVE product are expected to be of comparable quality). In regions and periods where ERS data are scarce due to the failure of the on-board storage (resulting in a temporal coverage below 15%), passive observations are used to fill these gaps even if the correlation and VOD threshold suggests the use of the ACTIVE product only.

In summary, the merging algorithm behind ESA CCI SM v2 COMBINED product is a ternary decision scheme that selects
either active-microwave-based retrievals alone, passive-microwave-based retrievals alone or an unweighted average of the two based on their mutual correlation and average VOD conditions at a particular time and location.



## 5 On the statistical optimality of data merging

While the ternary ESA CCI SM v2 decision tree algorithm described above has proven itself to be a robust way of merging soil moisture products from various satellites (Dorigo et al., 2015), it will hardly ever provide estimates that are optimal in a statistical sense. As known from (Generalized) Least Squares, deriving the best linear unbiased estimator for a measurand

from different simultaneous measurements of that measurand with supposedly different quality requires rigorous consideration of their individual errors and error correlations (Aitkin, 1935). Specifically, such an optimal estimate would be the weighted average of the individual measurements with the weights being derived from their error variances and covariances (Gelb, 1974). To understand this, consider an arbitrary number of $N$ simultaneous measurements of the measurand $y$, contained in the measurement vector $\boldsymbol{x}$ :

$\boldsymbol{x} = \mathbf{A} \cdot \boldsymbol{y} + \boldsymbol{\varepsilon}$          (1)

The $(N \times 2)$ design matrix $\mathbf{A}$ represents zeroth and first order (additive and multiplicative) systematic errors errors in $\boldsymbol{x}$ and the column vector $\boldsymbol{\varepsilon}$ represents independent (from $y$) additive Gaussian random errors in $\boldsymbol{x}$. The measurand vector $\boldsymbol{y} = (1 \quad y)^{\mathsf{T}}$ allows for the consideration of additive systematic errors. Note that (1) could be easily extended with higher-order systematic errors by extending the column dimension of the design matrix and the row dimension of the measurand vector, but not with

different types of random errors as Least Squares per definition allows for independent additive Gaussian noise only. In any case, (1) is the most commonly used error model for soil moisture data sets (Gruber et al., 2016b). The Least Squares solution, that is, the minimum random error variance estimate for $y$, is given as:

$$\boldsymbol{y} = (\mathbf{A}^{\mathsf{T}} \mathbf{P} \mathbf{A})^{-1} \mathbf{A}^{\mathsf{T}} \mathbf{P} \boldsymbol{x} \qquad (2)$$

where the weight matrix $\mathbf{P} = \mathbf{C}^{-1}$ is the inverse of the error covariance matrix whose diagonal elements are the error variances

of the measurements ($\sigma^2_{\varepsilon_i}$ with $i \in \boldsymbol{x}$) and whose off-diagonal elements are their error covariances ($\sigma_{\varepsilon_i, \varepsilon_j}$ with $i, j \in \boldsymbol{x}$ and $i \neq j$).

In practice, the success of (2) depends on the degree to which systematic errors and the error covariance matrix (i.e. $\mathbf{A}$ and $\mathbf{C}$) can be accurately estimated. Note, however, that even if only relative systematic differences between the measurements are known, (2) still provides a minimum random error variance estimate (up to the remaining unknown systematic component).

Recently, Gruber et al. (2017) proposed an implementation of (2) for merging active and passive microwave soil moisture retrievals, which utilises triple collocation analysis (TCA; Stoffelen, 1998; Gruber et al., 2016b) to estimate the input data uncertainties (i.e. diagonal elements of $\mathbf{C}$) and CDF-matching for a-priori correction of relative systematic differences. This merging scheme formed the basis for the ESA CCI SM v3 product.

## 6 ESA CCI SM v3

Gruber et al. (2017) showed that for a combination of ASCAT and AMSR-E retrievals a Least Squares merging scheme approach based on TCA outperforms the ternary merging scheme of ESA CCI SM v2. In the following sections we discuss



how the scheme was implemented and adapted for merging four active and seven passive input data sets into the ACTIVE, the PASSIVE, and the COMBINED ESA CCI SM v3 products (Dorigo et al., 2018a). The principal steps are illustrated in Figure 3. Note that the ESA CCI SM v3 algorithm continues to employ a two-stage merging scheme. That is, all active and passive data sets are first merged into the ACTIVE and the PASSIVE product, respectively, which are then further merged into the
COMBINED product.

## 6.1   Harmonisation of L2 data

The input data harmonisation is largely identical to that in ESA CCI SM v2 (see Section 4.1). ASCAT observations from MetOp-B, which were additionally included in v3, are treated as perfectly inter-calibrated with those from MetOp-A and arithmetically averaged on days and at locations where both satellites provide collocated measurements. SMOS, which was
additionally included in v3, is, as all the other products, rescaled (CDF-matched) against AMSR-E.

## 6.2   Uncertainty estimation for (harmonised) L2 data

Following Gruber et al. (2017), estimates of the random error variances (i.e. uncertainties) of the L2 data sets, required for the parametrisation of the employed Least Squares merging scheme, are obtained through TCA. TCA simultaneously estimates uncertainties of three spatially and temporally collocated data sets whose errors are required to be mutually uncorrelated. This
requirement is commonly assumed to be met when applying TCA to one active-microwave-based, one passive-microwave-based, and one land surface model based soil moisture data set (Scipal et al., 2008; Gruber et al., 2016b). Accordingly, uncertainties of all active and passive products (except for SMMR, which does not temporally overlap with any of the active data sets) are estimated as:

$$\sigma_{\varepsilon_a}^2 = \sigma_a^2 - \frac{\sigma_{a,p}\sigma_{a,m}}{\sigma_{p,m}}$$
$$\sigma_{\varepsilon_p}^2 = \sigma_p^2 - \frac{\sigma_{p,a}\sigma_{p,m}}{\sigma_{a,m}} \tag{3}$$

where the subscripts refer to the active ($a$), the passive ($p$) and the land surface model ($m$) time series; $\sigma_i^2$ is the temporal variance of data set $i$; and $\sigma_{i,j}$ is the temporal covariance between data sets $i$ and $j$ with $i,j \in [a,p,m]$. Uncertainty estimates for each active (passive) data set are obtained by applying (3) to that data set in combination with the respective passive (active) data set with the longest temporal overlap and GLDAS-Noah.

Note that the $\sigma_{\varepsilon_i}^2$ represent temporal mean data set uncertainties. Consequently, weights derived thereof (i.e. the $\mathbf{P}$ matrix in (2)) are average weights for the period for which TCA was applied, although actual retrieval uncertainties (of individual sensors) may change over time (see Section 9). For more details on TCA we refer to Gruber et al. (2016b). Note also that while errors of active and passive products are commonly assumed to be uncorrelated, significant correlations between the errors of different passive products may occur. Therefore, merging them into the PASSIVE product would require estimates of
these error correlations in order to properly parametrise the full error covariance matrix in (2). Gruber et al. (2016a) proposed a modification of TCA which potentially allows the estimation of such error correlations, but this method has not yet been





validated on a global scale and was found to be particularly susceptible to small sample sizes (sample sizes of products that are merged into the PASSIVE product are considered small in this context). Hence, lacking the ability to robustly estimate them, off-diagonal elements in the error covariance matrix are neglected in ESA CCI SM v3. The consequences of doing so are discussed in the next section and in Section 9.

## 6.3 Generation of the ACTIVE and the PASSIVE product

As in ESA CCI SM v2, the ACTIVE product is generated by concatenating the harmonised ERS and ASCAT time series because they do not have temporally overlapping observations which would allow for statistical merging. Consequently, their uncertainties, estimated from TCA, are merely appended to the product as auxiliary information and not used in the merging scheme.

Passive data sets are merged into the PASSIVE product as follows. Before October 2007 (i.e. before the launch of Coriolis, carrying WindSat), the low temporal coverage of the available sensors was assumed to render TCA-based error variance estimates too uncertain for a robust derivation of relative merging weights due to the susceptibility of TCA to small sample sizes (Zwieback et al., 2012). Consequently, SMMR, SSM/I, TMI and AMSR-E observations before October 2007 are concatenated in the same way as in ESA CCI SM v2, that is, by selecting the best available sensor at a particular time and location (see

Figure 2). Note that this was a relatively ad hoc albeit conservative assumption which has not yet been tested thoroughly but will be for future product versions (see Section 9).

All sensors available after this period (i.e. AMSR-E, WindSat, SMOS and AMSR2) are merged by employing the Least Squares estimator in (2) in the following manner: since the data sets are already harmonised, i.e. rescaled into a common climatology, the design matrix $\mathbf{A}$ is taken to be a ones column vector. The error covariance matrices $\mathbf{C}$ required to calculate

$\mathbf{P}$ (i.e. the relative weights for averaging the data sets) are constructed for each grid cell and for each merging period from the TCA-based uncertainty estimates of all sensors that are available during that particular period (see Figure 2). As mentioned above, error cross-correlations are neglected. That is, off-diagonal elements in $\mathbf{C}$ are held zero, which may lead to biases in the estimated weights in case errors of different passive data sets are significantly correlated (see Section 9). However, while this may reduce the efficiency (in uncertainty reduction) of the Least Squares estimator in several cases, it can not lead to

a substantial uncertainty increase (with respect to the individual L2 input products) because error correlations only pull the weights further towards the best product. If neglected, better products are still attributed with higher weights.

Note that the different sensors do not provide valid retrievals at every time step due to their orbit geometry and the L2 quality control (see Section 3). Consequently, if, during a particular merging period (see Figure 2), a data set with significantly larger uncertainties has a higher temporal measurement coverage than the others, simply merging all available observations at each

time step might result in a significantly larger overall uncertainty (of the merged time series) than that of the lower-uncertainty input time series alone. Therefore, to provide a trade-off between the best possible temporal measurement density and the lowest possible (average) uncertainty of merged time series, ESA CCI SM v3 imposes a minimum threshold for the cumulative weight of valid measurements available at a particular date, which has to exceed $1/2N$ where $N$ is the number of sensors in





orbit and operational during that merging period. If this threshold is not met, no soil moisture estimate is provided for that day. For more details on the choice and the implications of this threshold see Gruber et al. (2017).

## 6.4 Harmonisation of the ACTIVE and the PASSIVE product

As was the case with ESA CCI SM v2, in ESA CCI SM v3, before merging the ACTIVE and the PASSIVE products, their

climatologies are harmonised by rescaling them against GLDAS-Noah soil moisture simulations.

## 6.5 Uncertainty estimation for the (harmonised) ACTIVE and PASSIVE products

As mentioned in Section 6.2, TCA estimates represent the average retrieval uncertainties during the period in which TCA was applied. Since the uncertainties in the ACTIVE and PASSIVE products change significantly depending on which input data sets are used or merged at a particular point in time, uncertainties are estimated (from (3)) separately for all periods with

different sensor availabilities (see Figure 2). In addition to TCA-based uncertainty estimates, significance levels (p-values) of the Pearson correlation between the ACTIVE data set, the PASSIVE data set and GLDAS-Noah are calculated (separately for the same periods) in order to screen for unreliable TCA estimates (see the next section).

## 6.6 Generation of the COMBINED product

TCA estimates of soil moisture uncertainty are known to have limited reliability in certain regions such as deserts, high latitudes

or areas with dense vegetation (Dorigo et al., 2010; Al-Yaari et al., 2014). Using these estimates to parametrise the covariance matrix in (2) could thus significantly alter the integrity of the Least Squares estimator. Therefore, following the approach of Gruber et al. (2017), p-values are used to verify the reliability of TCA estimates and to fall back to the use of either active or passive retrievals alone, an unweighted average of the two, or to completely mask the grid cell during that period if uncertainty estimates and/or soil moisture retrievals are deemed unreliable. Specifically, the decision of whether to use the ACTIVE product

alone, the PASSIVE product alone, an unweighted average of the two, the Least Squares estimate, or to disregard the grid cell completely, is based on the relative p-value combination as illustrated in Table 2. If the Least Squares estimator is used, a minimum-weight threshold of 0.25 ($1/2N$ where $N = 2$, i.e. ACTIVE+PASSIVE) is again imposed on dates where only one of the data sets (ACTIVE or PASSIVE) provides a valid observation (see Section 6.3). More details and an evaluation of this classification scheme is provided in Gruber et al. (2017).

Figure 4 shows the relative weights during each merging period (see Figure 2) which are used for merging the ESA CCI SM v3 ACTIVE and PASSIVE products based on the TCA uncertainty estimates and the p-value mask. As a reference, the weight distribution amongst the ACTIVE and PASSIVE product in the ESA CCI SM v2 algorithm (only during the last merging period) and average VOD conditions at each location are given. The main apparent feature is that weight distributions in all merging periods largely follow VOD patterns. While the ESA CCI SM v2 algorithm was specifically designed to do so, the fact

that the uncertainty-based weights in the ESA CCI SM v3 algorithm do this as well (with a much better resolution) strengthens the evidence for the assumption that active products tend to perform better in more densely vegetated areas whereas passive





products tend to perform better in more sparsely vegetated regions. This behaviour forms the basis for the improved weight derivation in the ESA CCI SM v4 algorithm for regions where TCA estimates are deemed unreliable, which is discussed in the following section.

## 7 ESA CCI SM v4

5 Algorithmic changes which were implemented for generating the ESA CCI SM v4 products (Dorigo et al., 2018b) tackled two specific issues with the way in which the uncertainty estimates for the Least Squares merging are obtained in ESA CCI SM v3. First, the two-stage merging approach caused biases in the relative weights that are attributed to the ACTIVE and PASSIVE products during the different merging periods (see Figure 2). These biases resulted from the irregular temporal measurement availability of the individual L2 input data sets, which led to temporal uncertainty variations in the PASSIVE product during 10 the different merging periods depending on which sensors have valid observations and are merged together on a particular day. Such uncertainties variations cannot be accurately captured by the single uncertainty estimates used to merge the ACTIVE and PASSIVE products together. Second, even though seemingly robust, the p-value based ternary decision in areas where TCA estimates are deemed unreliable also resulted in biased (with respect to statistically optimal) weight estimates very similar to the biases in the ESA CCI SM v2 algorithm because it selects weights of 0, 0.5, or 1 irrespective of the actual data set 15 uncertainties (see Section 5). The following sections will describe the changes that have been implemented to address these issues. The resulting modified ESA CCI SM v4 algorithm is illustrated in Figure 5.

### 7.1 Direct merging of L2 observations into the COMBINED product

In ESA CCI SM v4, the COMBINED product is generated by directly merging L2 input data sets instead of the previously merged ACTIVE and PASSIVE products (as is the case in ESA CCI SM v2 and v3). This allows the estimation of temporally 20 dynamic relative merging weights for each individual sensor based on which sensors provide valid observations on a particular day. For this purpose, all L2 input data sets are first directly scaled against GLDAS-Noah to harmonise their climatology (as for the earlier versions, the SSM/I climatology is first replaced with that of AMSR-E). Uncertainties are then estimated (see below) for each individual product and used to construct error covariance matrices for all merging periods depending on the sensor availability during these periods (see Figure 2). Finally, the data sets are merged into the COMBINED product, again 25 using the minimum-weight threshold of $1/2N$ on dates where not all input products available in that merging period provide valid measurements (see Section 6.3).

### 7.2 VOD-based uncertainty estimation

As was shown in Figure 4, uncertainty estimates and hence merging weights largely follow VOD patterns. To obtain uncertainty estimates in regions where TCA estimates are not trusted, i.e. where not all three data sets used in TCA are significantly 30 correlated, an empirical polynomial regression approach that predicts uncertainties from average VOD conditions at a particular location was introduced. Specifically, a polynomial function is fitted between mean VOD (estimated from AMSR-E C-band





observations between 2002 and 2011) and TCA-based Signal-to-Noise Ratio (SNR) estimates using VOD and SNR tuples from all grid cells where TCA estimates are assumed to be reliable, i.e. where all three data sets are significantly ($p < 0.05$) correlated (see Figure 6). Regression coefficients are calculated separately for each L2 input product and used to predict their SNR levels ($\widehat{\mathrm{SNR}}_i$) from the mean VOD at grid cells ($i$) where the SNR could not be estimated from TCA:

$$\widehat{\mathrm{SNR}}_i = \sum_{j=0}^{k} a_j \cdot \overline{\mathrm{VOD}}^j \tag{4}$$

where $a_j$ are the polynomial coefficients and $k$ is the degree of the polynomial function. $k$ was chosen to be 3 for TMI and WindSat and 2 for all other sensors, which was empirically found to provide the best fit for the regression. Notice that regression coefficients are fitted between VOD and SNRs and not between VOD and uncertainties directly in order to account for varying signal variance across the grid cells that are used for the regression (Gruber et al., 2016b). SNRs are then converted

into uncertainties as:

$$\sigma_{\varepsilon_i}^2 = \frac{\sigma_i^2}{1 - \widehat{\mathrm{SNR}}_i} \tag{5}$$

The overshooting in the regression curve of TMI for high VOD values does not impact the final data product as grid cells with such high VOD values are masked out by the L2 quality control process. The overshooting of WindSat for low VOD values affects a few grid cells in very dry regions and can not be avoided by changing the polynomial order as this would lead to

overshooting in the more relevant VOD regions. SNR values at different grid cells and for particular VOD ranges sometimes show a significant variability around the corresponding estimate of the regression, which directly translates to uncertainties in the weight estimates that are used for the Least Squares merging. However, these uncertainties are assumed to be, on average, lower than the bias introduced by the p-value based ternary decision of a weight of either 0, 0.5 or 1 as adopted in ESA CCI SM v2 and v3.

Figure 7 shows the combined TCA and VOD regression based global SNR maps which are ultimately used to derive the merging weights for (2). Patterns generally follow common understanding. SNRs of active sensors are higher in more densely vegetated regions whereas SNRs of passive sensors are higher in more sparsely vegetated areas (Dorigo et al., 2010; Liu et al., 2011, 2012; van der Schalie et al., 2018). SNRs of passive sensors largely depend on the microwave frequency (Liu et al., 2012; Parinussa et al., 2011, 2012). AMSR-E and AMSR2 (both C-band) SNRs are largely comparable and, in general, are higher

than those for the higher-frequency (Ku-band) of SSM/I. SMOS (L-band) SNRs are, on average, relatively high and show a lower spatial variability as its longer wavelength makes the observations less sensitive to vegetation variations.

Note that, as is the case in ESA CCI SM v3, SSM/I and TMI retrievals are never merged together or merged with AMSR-E, i.e. SSM/I data are only used at high-latitudes where TMI data are not available, and neither of the two is used after AMSR-E becomes available (see Section 6.3). Nevertheless, their uncertainties are in many areas comparable with each other and with

those of the other sensors, suggesting that they might add valuable information when included in the Least Squares merging scheme, which will be considered for future product versions (see Section 9).



### 7.3 P-value based quality control

In ESA CCI SM v4, for the generation of both the PASSIVE and the COMBINED product, correlation significance levels are used to completely mask out individual L2 input products that are deemed unreliable at a particular location and during a particular merging period (in cases where more than one product is available for merging). For this purpose, the p-value mask that is used in the ESA CCI SM v3 product (see Section 6.6) was modified as shown in Table 3. All measurements from the target satellite product that is being tested for reliability are masked out if they do not correlate significantly with both soil moisture estimates from GLDAS-Noah and the measurements from the second satellite product used for TCA, or if they correlate significantly with the reference satellite product but not with the model time series.

The rationale behind the latter is that potential non-zero error correlations, arising, for example, from uncorrected vegetation variations (Zwieback et al., 2018), may lead to spurious correlations between the two products even though they do not contain useful soil moisture information. Note, however, that the decisions in the p-value mask were empirically tuned to lead to a good performance (of the merged products) in terms of correlation against the ERA-Land soil moisture product. Consequently, decisions that are based on significance levels of the correlation against GLDAS-Noah may be questionable since the two models are most likely not fully independent. This issue is currently under investigation and will be addressed in future product versions (see Section 9).

## 8 Product evaluation

The previous sections provided a methodological review of the merging algorithm behind the ESA CCI SM (and Copernicus C3S) products. Even though this is not a validation paper, this section shall provide an overview of the performance evolution of the presented product versions, i.e. ESA CCI SM v2, v3, and v4. To this end, both absolute and anomaly time series (calculated by removing seasonal dynamics which are estimated by applying a 35-day moving average window) of the ACTIVE, PASSIVE, and COMBINED data sets from the latest public release of each product version (i.e. v02.2, v03.3, and v04.4) are correlated against globally distributed in situ soil moisture observations from the International Soil Moisture Network (ISMN; Dorigo et al., 2011a, b). Only in situ measurements that are flagged "good" by the ISMN internal quality control (Dorigo et al., 2013) are used for the comparison. Unreliable ESA CCI SM soil moisture estimates are masked out as described in the previous sections. Products are evaluated from October 2007 onwards as significant improvements are mainly expected after this date due to the use of multiple passive satellites within the merging scheme (see Section 6) and the improved temporal data coverage of both the ESA CCI SM products and the ISMN stations available for validation.

The majority of ISMN stations are distributed over large areas and most ESA CCI SM grid cells contain only single measurement stations. Direct comparisons (i.e. relative correlation coefficients) are therefore affected by significant upscaling errors (in addition to in situ sensor measurement errors; Miralles et al., 2010; Gruber et al., 2013). TCA potentially allows to avoid this influence by directly estimating correlation coefficients with respect to the unknown "true" soil moisture signal (McColl et al., 2014). Chen et al. (2017) showed, that these TCA-based correlation coefficients are independent of in situ sensor and representativeness errors. However, TCA requires the errors of the data sets to which it is applied to be mutually independent. Usually,





any combination of in situ soil moisture measurements, active-microwave-based soil moisture retrievals, passive-microwave-based soil moisture retrievals, and modelled soil moisture estimates is expected to fulfil this requirement (see Section 6.2; Gruber et al., 2016b), but since the ESA CCI SM COMBINED product is generated by using the latter three data sources, no data triplet that meets TCA assumptions can be found to evaluate this product.

Here we circumvent this issue through the application of Bayes theorem (Efron, 2013). To this end, we acquire prior estimates of ISMN sensor plus representativeness errors in terms of their correlation with respect to "true" soil moisture signal at the satellite scale ($R_i$) by applying TCA to the ISMN stations together with the ESA CCI SM ACTIVE and PASSIVE products (Chen et al., 2017):

$$R_i = \sqrt{\frac{\sigma_{i,a}\sigma_{i,p}}{\sigma_{i,i}\sigma_{a,p}}} \tag{6}$$

where $\sigma$ denotes the covariance between data sets and the subscripts denote the ISMN stations ($i$) and the ACTIVE ($a$) and PASSIVE ($p$) products. This prior information now allows to derive estimates of the correlation of the different ESA CCI SM products against the "truth" ($R_e$) from their relative Pearson correlation against the ISMN stations ($R_{e,i}$) through Bayesian inference:

$$R_e = \frac{R_{e,i}}{R_i} \tag{7}$$

In other words, Eq. (7) corrects the Pearson correlation between ISMN stations and the ESA CCI SM products for the impact of ISMN sensor and representativeness errors. An analytical proof for the relation in Eq. (7) can be found by using the general definitions of the Pearson correlation coefficient and the TCA-based correlation against the unknown "truth" (McColl et al., 2014; Gruber et al., 2016b):

$$R_{x,y} = \frac{\sigma_{x,y}}{\sigma_x\sigma_y} \qquad R_x = \sqrt{\frac{\sigma_{x,y}\sigma_{x,z}}{\sigma_{x,x}\sigma_{y,z}}} \qquad R_y = \sqrt{\frac{\sigma_{x,y}\sigma_{y,z}}{\sigma_{y,y}\sigma_{x,z}}} \tag{8}$$

Estimates of $R_i$ are calculated for both absolute and anomaly time series of all ISMN stations using the maximum possible temporal overlap with the ESA CCI SM ACTIVE and PASSIVE products. $R_e$ estimates are then calculated for absolute and anomaly time series of each ESA CCI SM product (i.e. ACTIVE, PASSIVE, and COMBINED version v02.2, v03.3, v04.4) for each merging period after October 2007 (see Figure 2) using only dates where all three product versions have valid measurements. Estimates are masked out at locations where not all products are significantly correlated (p< 0.05) or have less

then 100 collocated measurements (Gruber et al., 2016b). $R_e$ estimates that exceed unity (which may occur due to statistical sampling errors; Gruber et al., 2018) are set to one. Figure 8 shows the locations of all 1056 ISMN stations where valid $R_e$ estimates could be obtained (in any of the four considered merging periods).

    Spatial statistics of the estimated correlations ($R_e$) are shown in Figure 9. Clear improvements for increasing ESA CCI SM product version are visible for the PASSIVE and the COMBINED products in almost all merging periods, both for absolute soil

moisture time series and for anomalies. No significant anomaly correlations, and significant absolute correlations from only 4 sites are available for the PASSIVE product in the merging period between October 2011 and June 2012, which is associated with the low data coverage of WindSat and SMOS that are used in this period, and the predominant frozen conditions during



this time of the year, which lead to further masking of most data points. Only slight, non-significant changes are visible for the ACTIVE product, which is expected because only a single sensor is used and no statistical merging is applied that would be affected by changes between product versions. Also the inclusion of MetOp-B observations as of ESA CCI SM v3 is unlikely to influence the results as only dates where all three ESA CCI SM product versions provide valid observations are considered in

the analysis. Therefore, apparent changes originate mainly from differences in the L2 soil moisture retrieval algorithm version that has been used for ASCAT (see Table 1), more specifically from model parameter updates due to time series extension. For a comprehensive summary of validation studies for the various ESA CCI SM product versions we refer the reader to Dorigo et al. (2015, 2017).

## 9   Conclusions

This paper reviews the evolution of the merging methodology behind the European Space Agency's Climate Change Initiative for Soil Moisture (ESA CCI SM) Climate Data Records (CDRs). The ESA CCI SM algorithm generates consistent, quality-controlled, long-term (1978 - 2018) soil moisture CDRs by harmonising and merging soil moisture retrievals from multiple satellites into (i) an active-microwave-based only (ACTIVE), (ii) a passive-microwave-based only (PASSIVE), and a combined active-passive (COMBINED) product, which are sampled to daily global images on a 0.25 degree regular grid. Since the

first product release in 2012, the merging methodology has undergone substantial improvements which have so far not been thoroughly reported in the scientific literature. This paper reviews and discusses the science behind the three major ESA CCI SM merging algorithm versions:

– ESA CCI SM v2, which was used for all product releases between 2012 and 2016. This algorithm merges active and passive soil moisture retrievals by selecting either one of them alone or by computing the unweighted average of both

based on their mutual correlation and average Vegetation Optical Depth (VOD) conditions at a given location and time period.

– ESA CCI SM v3, which was released early 2017 and extended at the end of 2017, and used for the near-real-time Copernicus Climate Change Service (C3S) Soil Moisture CDR production up to version v201801. This algorithm uses a Weighted Least Squares based merging scheme, which is parametrised by triple collocation analysis (TCA) based

uncertainty estimates and uses correlation significance levels (p-values) to fall back to a ternary decision scheme (active-only, passive-only, or an unweighted average) at grid cells and/or during time periods where TCA-based uncertainty estimates are deemed unreliable.

– ESA CCI SM v4, which was used to generate the product releases at the beginning and the end of 2018, and has been used for the C3S Soil Moisture CDR production since version v201812. This algorithm introduced a VOD-based polynomial

regression to obtain global uncertainty estimates for all products (i.e. also in regions where TCA-based estimates are not reliable) and directly merges all active and passive L2 soil moisture retrievals into the COMBINED product (i.e., no longer the previously merged ACTIVE and PASSIVE products).



Harmonising soil moisture retrievals from active and passive microwave measurements from instruments which (i) operate at different wavelengths, polarisations and incidence angles; (ii) have diverging spatial, temporal and radiometric resolution; and (iii) are hardly ever well collocated in space and time is a heavily ill-posed problem. The ESA CCI SM merging algorithm is hence subject to continuous research and development. In the following, we summarise known issues that are currently under
investigation and highlight improvements that are expected to be implemented in the next algorithm version (v5), which is foreseen to be released in 2019.

- L2 data usage

    - Soil moisture retrievals from SMAP (Entekhabi et al., 2010) will be integrated into the next algorithm version v5. SMAP retrievals are expected to significantly enhance the ESA CCI SM performance from 2015 onwards due to
its long wavelength (L-band) and remarkably high radiometric accuracy (Chen et al., 2018).

    - SSM/I and TMI data are not yet fully integrated. At mid-latitudes, SSM/I data was disregarded in favour of TMI, and both products were cut off after the launch of the presumed better AMSR-E, because their low temporal coverage was assumed to render TCA-based uncertainty estimates (required for the Least Squares merging scheme) too unreliable (see Section 6.3). Nonetheless, their estimated uncertainties (see Figure 7) suggest potentially useful
complementary information even in the presence of the more recent missions.

    - In ESA CCI SM v2, WindSat is merely used for bridging the gap between the failure of AMSR-E and the launch of AMSR2. For this reason, WindSat data were only retrieved until mid 2012. However, due to L1 data availability issues, WindSat retrievals have not been extended since even though uncertainty estimates for WindSat (see Figure 7) suggest that more recent retrievals may benefit the ESA CCI SM product when integrated in the Least Squares
merging scheme.

    - All ESA CCI SM products are sampled on a 0.25 degree regular grid and incorporate only L2 retrievals from sensors that operate at a comparable resolution. However, high-resolution soil moisture retrievals from Synthetic Aperture Radar (SAR) instruments, in particular Envisat ASAR and Sentinel-1, are expected to provide useful complementary information either when upscaled to coarse-resolution or for downscaling the ESA CCI SM products.

- Data harmonisation

    - In current and previous product versions, AMSR-E and AMSR2 retrievals are treated as if they were perfectly inter-calibrated, that is, no harmonisation between the two is applied. However, visual time series inspections as well as preliminary studies suggest that remaining biases are present which should be removed before merging (Parinussa et al., 2015). The same may be the case for MetOp-A and MetOp-B ASCAT retrievals, even though no
significant discrepancies have been found yet.

    - The CDF-matching, which is used for harmonising L2 product climatologies, implicitly assumes that the considered data sets have an identical signal-to-noise ratio, which is hardly ever the case (see Figure 7). Therefore, rescaling





coefficients will most certainly be biased. TCA may provide an alternative approach for estimating optimal (in a Least Squares sense) scaling coefficients (Yilmaz and Crow, 2013).

– For the sake of consistency, L2 soil moisture estimates in all ESA CCI SM product versions are retrieved using the WARP algorithm for active microwave measurements and the LPRM algorithm for passive microwave measurements (see Section 3). The selection of these two algorithms was based on an extensive Round Robin comparison between various retrieval models (Gruber et al., 2014; Mittelbach et al., 2014). However, these choices may be worth reassessing especially due to the availability of new SMOS and SMAP products (O'Neill et al., 2018; Fernandez-Moran et al., 2017; Entekhabi et al., 2010).

– Uncertainty estimation

– In the current merging scheme (v4), uncertainties, and hence relative merging weights, are assumed to be (locally) stationary. That is, they are held constant during the entire time period for which TCA is applied. However, given their strong link with vegetation density, actual uncertainties are expected to vary significantly between seasons or with land cover change. Consequently, the estimation of non-stationary uncertainties could provide more accurate relative weightings on an intra-annual basis and thus a more efficient uncertainty reduction upon merging. Such time-variant uncertainty estimation, realized by decomposing the satellite time series into different frequency components and merging them separately (Draper and Reichle, 2015; Su and Ryu, 2015), is currently under investigation and foreseen to be integrated in a future release of the ESA CCI SM data set.

– The statistical merging that was introduced in ESA CCI SM v3 is a *Weighted* Least Squares implementation, which neglects possible error correlations across products. While such correlations are very likely to exist between the errors of the passive products they are usually not expected between errors of active and passive products, although the latter may be introduced by vegetation dynamics that are not completely removed in the retrieval (Gruber et al., 2016a; Zwieback et al., 2018) or imposed by the non-linear nature of the CDF-matching (see below). However, unless the relative uncertainties of the merged products (with correlated errors) are diverging by several orders of magnitude, non-zero error correlations will only cause sub-optimal and not significantly incorrect relative weighting. Nonetheless, if existing error cross-correlations could be estimated and considered in a *Generalized* Least Squares fashion (i.e. parametrising the currently neglected off-diagonal elements of the error covariance matrices) this could again lead to a significant performance improvement of the ESA CCI SM products. One option to do so could be so-called Extended Collocation Analysis (ECA; Gruber et al., 2016a), which is currently the only potentially available method for estimating error correlations between large-scale soil moisture products. However, the method has not yet been validated on a global scale and has been found to be particularly susceptible to small sample sizes, although this issue is expected to be mitigated by the progressively increasing data coverage of currently available missions.

– The polynomial regression for predicting uncertainties from VOD, which was introduced in ESA CCI SM v4, is based on long-term average C-band VOD estimates retrieved from AMSR-E. However, the functional relationship



between uncertainties and VOD may be different for the Ku-band retrievals from SSM/I, for the X-band retrievals from TMI and AMSR-E, and for the L-band retrievals from SMOS and SMAP, especially when considering their intra-annual variability. Therefore, VOD-based uncertainty predictions for individual sensors may be more accurate when obtained from a regression with VOD estimates in their respective frequency band and/or in a temporally dynamic manner.

– The p-value mask for excluding individual data sets that was introduced in ESA CCI SM v4 was implemented on a relatively conservative ad hoc basis and is pending a more thorough evaluation and refinement. Considering, for example, absolute or relative signal-to-noise ratios and/or relative weight differences may help to better balance temporal measurement density and data quality.

– Model dependency

– So far, climatologies are harmonised (for the COMBINED product) by CDF-matching individual products against the GLDAS-Noah land surface model. This may be particularly problematic for trend analysis because such rescaling imposes any natural or spurious trends existing in the model to a certain degree on to the harmonised ESA CCI SM product. Moreover, the non-linear nature of the CDF-matching may introduce spurious error correlations, which could be problematic for TCA (see above) but also when evaluating the ESA CCI SM data set against other land surface models such as ERA-Interim/Land or MERRA2, which hampers a comprehensive large-scale validation of the product.

– Apart from serving as scaling reference, GLDAS-Noah is also used as third data set to complement the data triplet used for TCA. This, per se, would not introduce spurious correlations between the merged ESA CCI SM product and the model because - in theory - each data set merely serves as independent "instrument" to isolate the individual error variabilities from the total variabilities present in each product (Su et al., 2014). However, this isolation is realised by using the jointly observed variability from the three products as reference for the "true" soil moisture variability (hence the requirement of uncorrelated errors) to derive the individual error variances as deviations from this jointly observed "true" soil moisture signal. Consequently, mismatches in the spatial representation (i.e. horizontal and vertical resolution) and temporal collocation may cause real soil moisture signals that are not captured by all three data sets (such as precipitation events not present in the model forcing) or signals that are seen by the satellite data sets but not represented in the land surface model (such as irrigation; Brocca et al., 2018) to be interpreted as representativeness errors or - looking from a different angle - as spurious error correlations (Miralles et al., 2010; Vogelzang and Stoffelen, 2012; Gruber et al., 2013, 2016b). This could again lead to biases in the estimated uncertainties and hence merging weights. It is therefore desirable to avoid the use of a land surface model not only in the harmonisation process but also in TCA, e.g., by replacing it with lagged version of the satellite products (Su et al., 2014).



## 10   Data availability

The soil moisture CDRs produced within the ESA CCI SM are freely available upon registration at http://www.esa-soilmoisture-cci.org/ or at the Centre for Environmental Data Analysis (CEDA) via http://dx.doi.org/10.5285/3729b3fbbb434930bf65d82f9b00111c (ESA CCI SM v2; Wagner et al., 2018), http://dx.doi.org/10.5285/b810601740bd4848b0d7965e6d83d26c (ESA CCI SM v3; Dorigo et al., 2018a), and http://dx.doi.org/10.5285/3a8a94c3fa464d68b6d70df291afd457 (ESA CCI SM v4; Dorigo et al., 2018b). ISMN data are freely available upon registration at https://ismn.geo.tuwien.ac.at/.

*Competing interests.*   The authors declare that they have no conflict of interest.

*Acknowledgements.*   This study was carried out within the eartH2Observe project (European Union's Seventh Framework Programme, Grant Agreement No. 603608) and ESA's Climate Change Initiative (CCI) for soil moisture (Contract No. 4000104814/11/I-NB), and supported by the KU Leuven C1 internal fund C14/16/045. The methods reported on within this paper are further integrated within the production system providing TCDR/ICDR to the Copernicus Climate Change Service (C3S) implemented by ECMWF on behalf of the European Commission. We thank the ISMN data providers for sharing their data with the community. Analyses in this paper are based on data from the following networks: AMMA-CATCH (http://www.amma-catch.org/; Pellarin et al., 2009), ARM (http://www.arm.gov/), AWDN (http://www.hprcc.unl.edu/awdn.php), BNZ-LTER (http://www.lter.uaf.edu/), CARBOAFRICA (http://dx.doi.org/10.7167/2013/297973; Ardö, 2012), COSMOS (http://coSMOS.hwr.arizona.edu/; Zreda et al., 2012), CTP_SMTMN (http://dam.itpcas.ac.cn/rs/?q=data#CTP-SMTMN; Yang et al., 2013) , DAHRA (http://ign.ku.dk/earthobservation/research/document4/CaLM/; Tagesson et al., 2015), FLUXNET-AMERIFLUX (http://ameriflux.lbl.gov/), FMI (http://fmiarc.fmi.fi/), FR_Aqui, GTK, HOBE (http://www.hobe.dk/; Bircher et al., 2012), ICN (http://www.isws.illinois.edu/warm/; Hollinger and Isard, 1994), IIT_KANPUR (http://www.iitk.ac.in/), MAQU (Su et al., 2011), MOL-RAO (http://www.dwd.de/mol), ORACLE (http://gisoracle.irstea.fr/?lang=en;https://bdoh.irstea.fr/ORACLE/), OZNET (http://www.oznet.org.au/; Smith et al., 2012), PBO_H2O (http://xenon.colorado.edu/portal/index.php?product=soil_moisture; Larson et al., 2008), REMEDHUS (http://campus.usal.es/~hidrus/, RISMA (http://aafc.fieldvision.ca/; Ojo et al., 2015), RSMN (http://assimo.meteoromania.ro/, SCAN (http://www.wcc.nrcs.usda.gov/), SMOSMANIA (http://www.hymex.org/; Albergel et al., 2008), SNOTEL (http://www.wcc.nrcs.usda.gov/; Leavesley et al., 2008), SOILSCAPE (http://soilscape.usc.edu/; Moghaddam et al., 2010), SWEX_POLAND (Marczewski et al., 2010), TERENO (http://teodoor.icg.kfa-juelich.de/; Zacharias et al., 2011), UDC_SMOS (http://www.geographie.uni-muenchen.de/department/fiona/forschung/projekte/index.php?projekt_id=103; Schlenz et al., 2012), UMBRIA (http://www.cfumbria.it/; Brocca et al., 2011), UMSUOL (http://www.arpa.emr.it/sim/), USCRN (http://www.ncdc.noaa.gov/crn/; Bell et al., 2013), USDA-ARS (https://www.ars.usda.gov/; Jackson et al., 2010), and WSMN (http://www.aber.ac.uk/wsmn/).



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





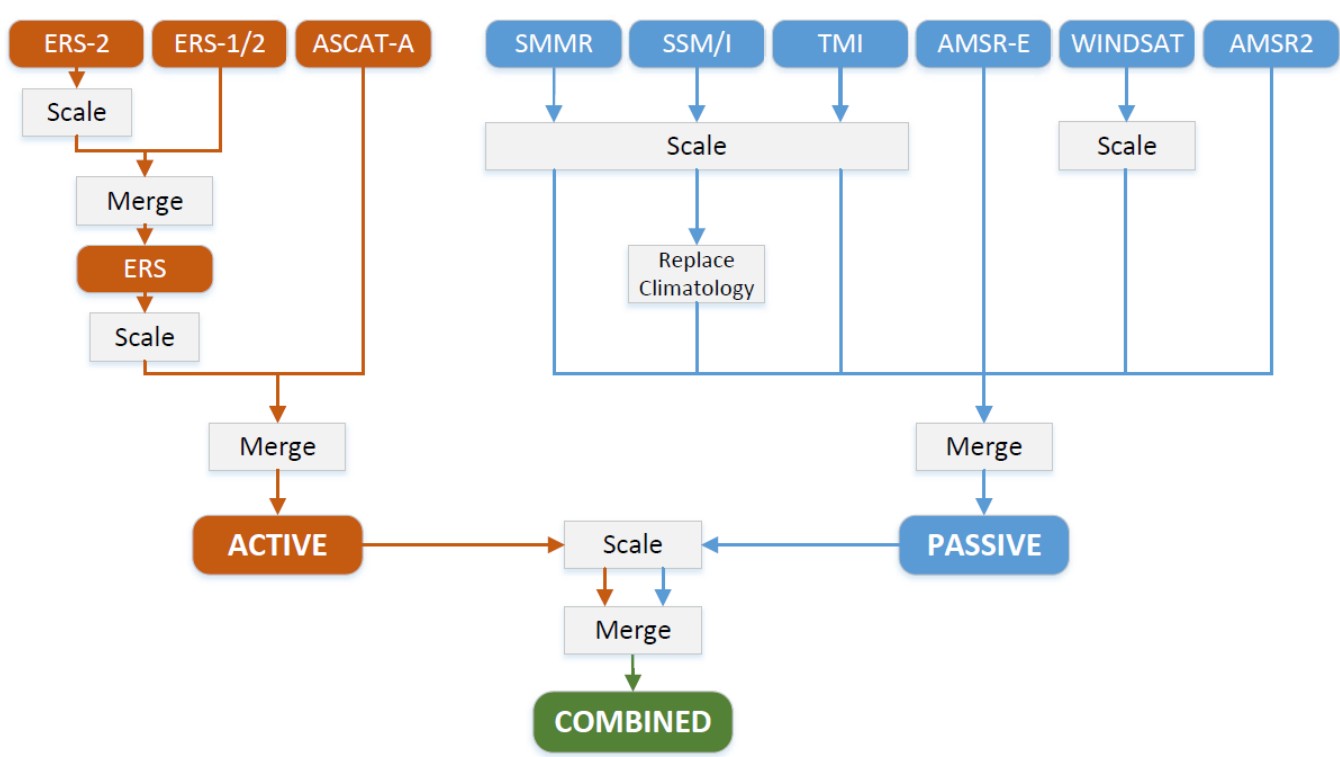

**Figure 1.** Merging scheme of the ESA CCI SM v2 algorithm.



**Figure 2.** Merging periods and sensor selection of the ACTIVE (top), PASSIVE (middle), and COMBINED (bottom) ESA CCI SM v2, v3, and v4 products.





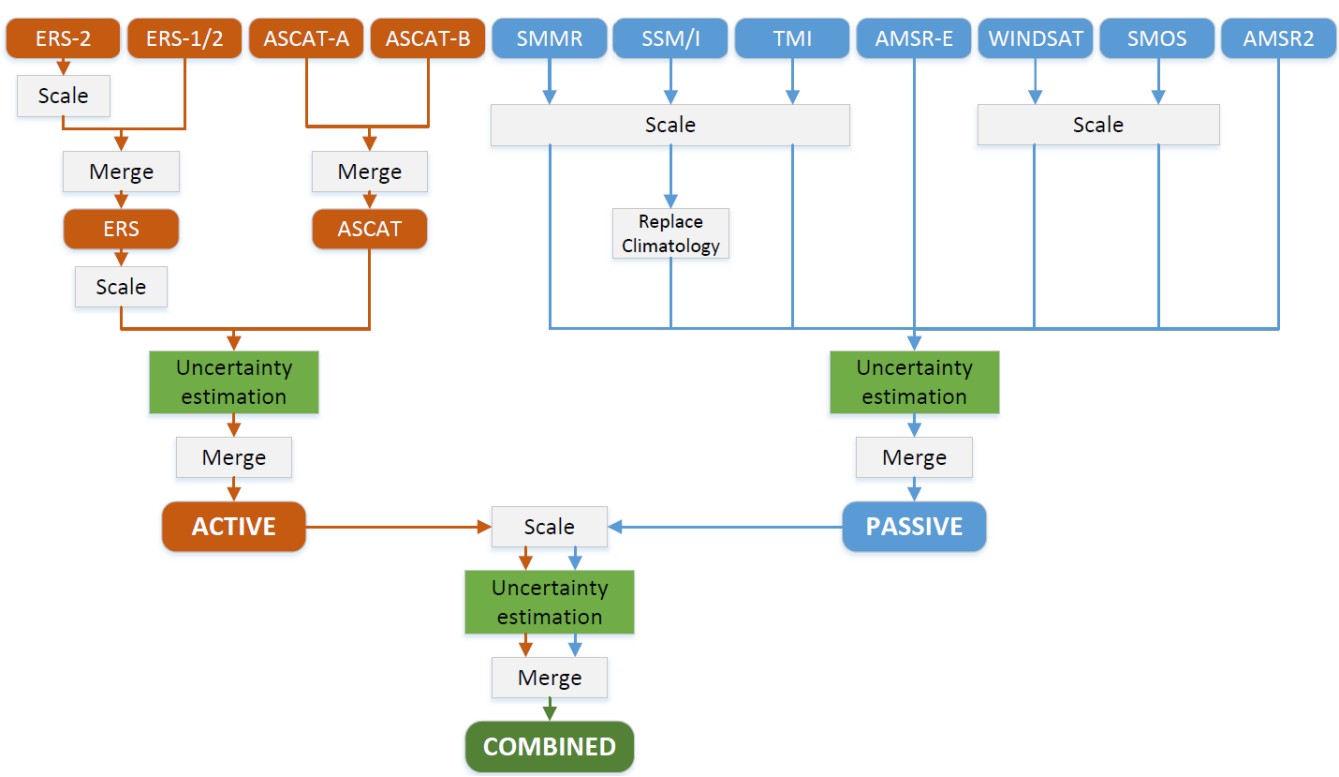

**Figure 3.** Merging scheme of the ESA CCI SM v3 algorithm.





**Figure 4.** Weights for merging the ACTIVE and PASSIVE products in the ESA CCI SM v3 algorithm (all sensor periods, top four rows), weights for merging the ACTIVE and PASSIVE products in the ESA CCI SM v2 algorithm (latest period, bottom left), and average VOD derived from AMSR-E (bottom right).



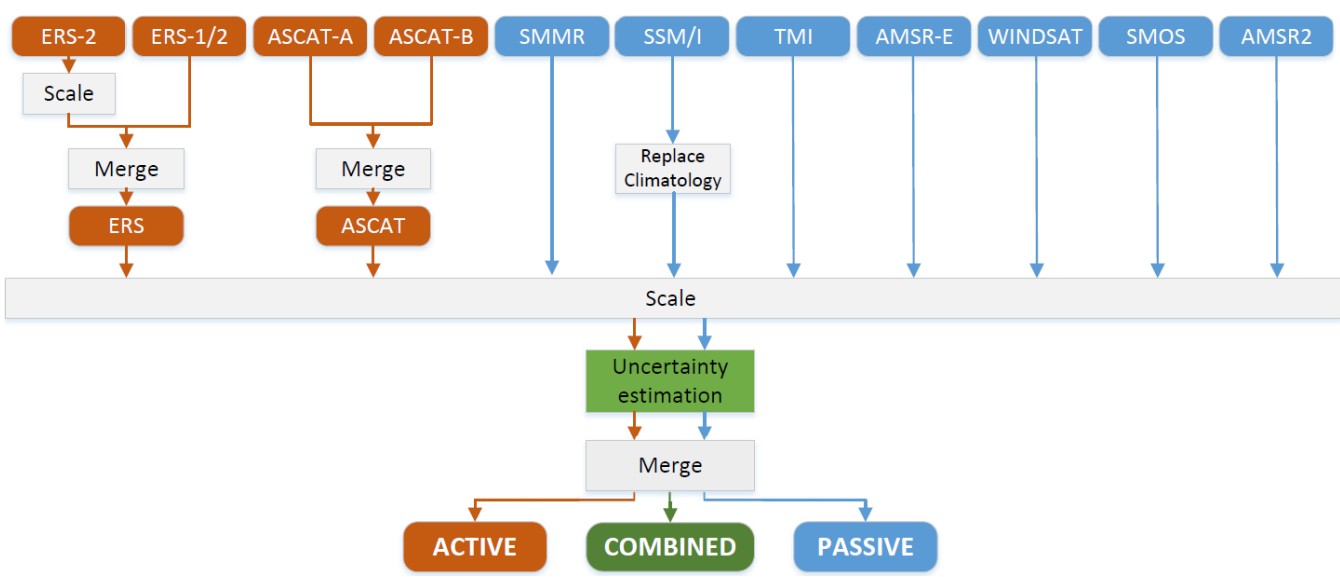

**Figure 5.** Merging scheme of the ESA CCI SM v4 algorithm.



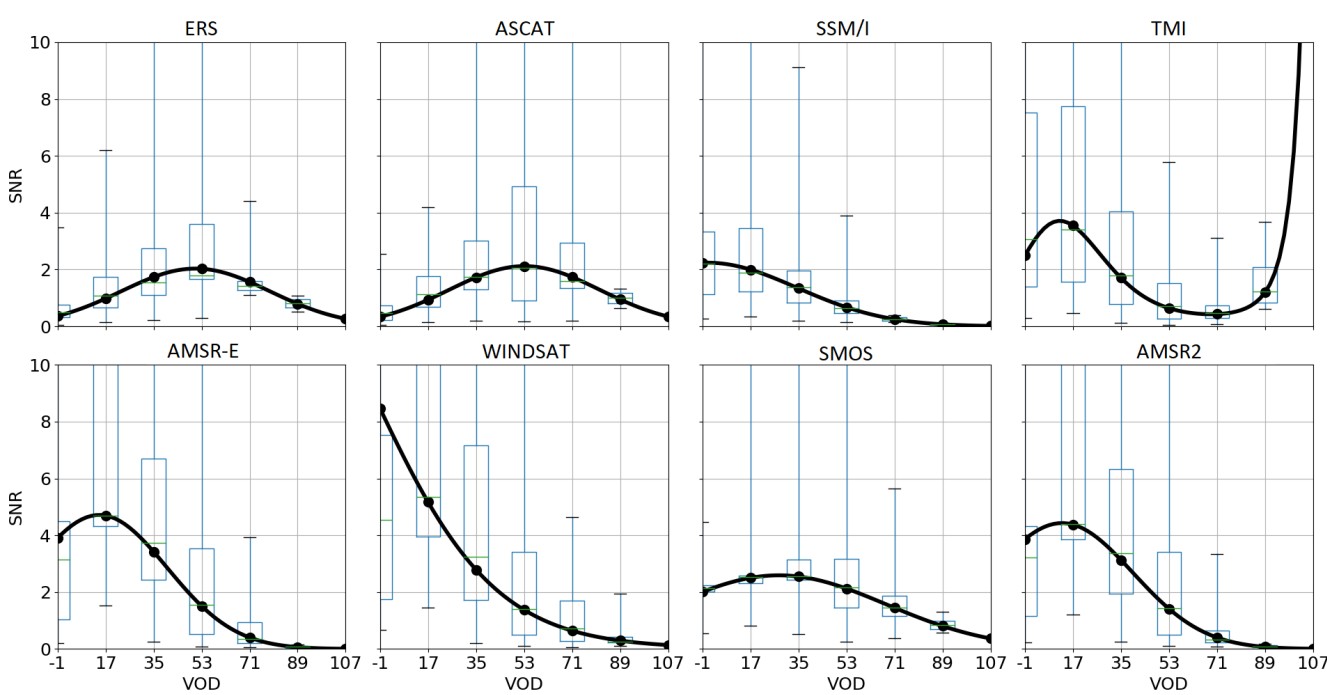

**Figure 6.** Regression functions between VOD and SNRs (in decibel units) of all L2 products. Boxplots show the median, the inter-quartile-range and the 5 and 95 percentiles, respectively.





**Figure 7.** SNR (in decibel units) of all L2 input products with uncertainties estimated from TCA and the VOD regression.





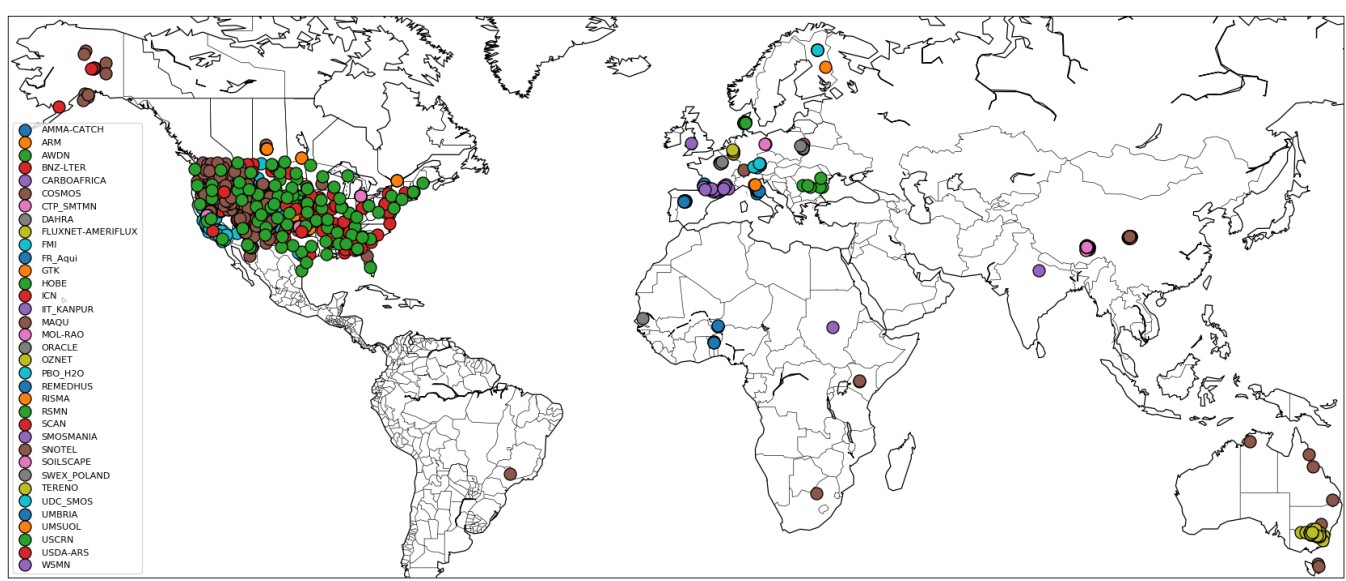

**Figure 8.** Locations of the ISMN stations used for product evaluation. Colors represent different measurement networks.





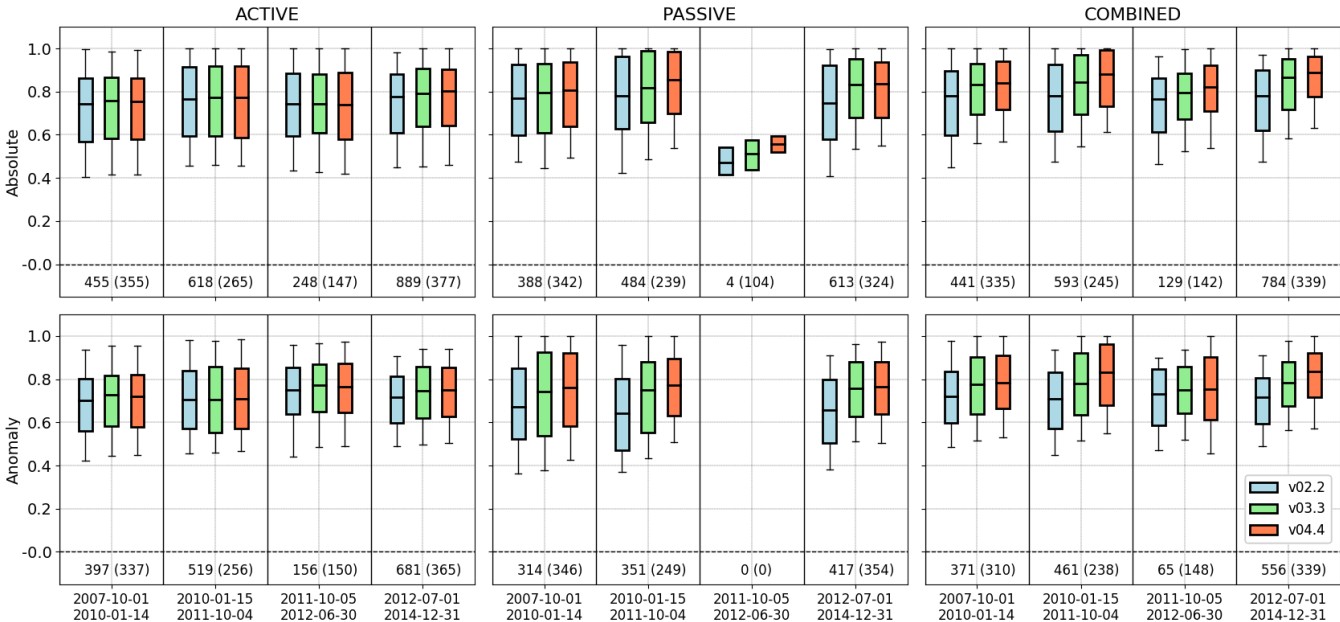

**Figure 9.** Boxplots of correlations against the unknown "truth" of measurements from the ACTIVE (left), PASSIVE (middle) and COMBINED (right) products, both for absolute time series (top) and soil moisture anomalies (bottom). Box colors refer to the ESA CCI SM product versions v02.2, v03.3, and v04.4. The x-axes represent different merging periods (see Figure 2). Boxes represent the median and IQR, and whiskers the 10th and 90th percentiles of significant correlations ($p < 0.05$) over all stations where at least 100 collocated measurements are available. The number of stations available for calculating the correlation statistics for a particular product and time period is shown below the zero line. The number in brackets shows the average number of collocated measurements available at each station for calculating correlation coefficients.



**Table 1.** Instrument characteristics for the products that are merged into the ESA CCI SM data sets (modified from Chung et al. (2018a)).

| | Passive microwave products | | | | | | | Active microwave products | | | |
|---|---|---|---|---|---|---|---|---|---|---|---|
| | SMMR | SSM/I | TMI | AMSR-E | Windsat | SMOS | AMSR2 | AMI-WS | AMI-WS | ASCAT-A | ASCAT-B |
| Platform | Nimbus 7 | DMSP | TRMM | Aqua | Coriolis | SMOS | GCOM-W1 | ERS1/2 | ERS2 | MetOp-A | MetOp-B |
| Time period | Jan 1979 – Aug 1987 | Sep 1987 – Dec 2007 | Jan 1998 – Dec 2013 | Jul 2002 – Oct 2011 | Oct 2007 – Jul 2012 | Jan 2010 – June 2018 | July 2012 – June 2018 | Jul 1991 – Dec 2006 | May 1997 – Feb 2007 | Jan 2007 – June 2018 | Nov 2012 – June 2018 |
| Algorithm version* | LPRM v05 LPRM v05 LPRM v05 | LPRM v05 LPRM v05 LPRM v05 | LPRM v05 LPRM v05 LPRM v05 | LPRM v05 LPRM v06 LPRM v06 | LPRM v05 LPRM v06 LPRM v05 | - LPRM v06 LPRM v06 | LPRM v05 LPRM v06 LPRM v06 | WARP 5.5 WARP 5.5 WARP 5.5 | WARP 5.4 WARP 5.4 WARP 5.4 | WARP 5.5 WARP 5.6 WARP 5.8 | - WARP 5.6 WARP 5.8 |
| Channel used [GHz] | 6.6 | 19.3 | 10.7 | 6.9/10.7 | 6.8/10.7 | 1.4 | 6.9/10.6 | 5.3 | 5.3 | 5.3 | 5.3 |
| Spatial resolution [km²] | 150 × 150 | 69 × 43 | 59 × 36 | 76 × 44 | 25 x 35 | 40 x 40 | 35 x 62 | 50 × 50 | 25 x 25 | 25 × 25 | 25 × 25 |
| Spatial coverage | Global | Global | N37° – S37° | Global | Global | Global | Global | Global | Global | Global | Global |
| Swath width [km] | 780 | 1400 | 780 / 897 | 1445 | 1025 | 600 | 1450 | 500 | 500 | 2 x 550 | 2 x 550 |
| Equatorial crossing | 0:00 (Dsc.) | 06:30 (Dsc.) | Varies | 01:30 (Dsc.) | 6:03 (Dsc.) | 6:00 (Asc.) | 01:31 (Dsc.) | 9:30 (Asc.) 10:30 (Dsc.) | 9:30 (Asc.) 10:30 (Dsc.) | 9:30 (Asc.) 10:30 (Dsc.) | 9:30 (Asc.) 10:30 (Dsc.) |
| Unit | m³m⁻³ | m³m⁻³ | m³m⁻³ | m³m⁻³ | m³m⁻³ | m³m⁻³ | m³m⁻³ | % sat. | % sat. | % sat. | % sat. |

*\* L2 retrieval algorithm versions refer to those used for ESA CCI SM v2 (top), v3 (middle), and v4 (bottom). SMOS and ASCAT-B data were not included in ESA CCI SM v2.*



**Table 2.** Merging scheme based on the one-tailed p-vale for the correlation between active (a), passive (p) and modelled (m) soil moisture with a 0.05 significance level (modified from Gruber et al. (2017)).

| p < 0.05? (0: no, 1: yes) | | | decision |
|---|---|---|---|
| a - m | p - m | a - p | |
| 0 | 0 | 0 | disregard pixel |
| 0 | 1 | 0 | passive only |
| 0 | 1 | 1 | |
| 1 | 0 | 0 | active only |
| 1 | 0 | 1 | |
| 1 | 1 | 0 | arithmetic mean |
| 0 | 0 | 1 | |
| 1 | 1 | 1 | Least Squares |





**Table 3.** Merging scheme based on the one-tailed p-vale for the correlation between the model (m), the reference satellite (r) the target satellite (t) soil moisture with a 0.05 significance level.

| p < 0.05? (0: no, 1: yes) | | | decision |
|---|---|---|---|
| m - r | m - t | r - t | |
| 0 | 0 | 0 | disregard t |
| 1 | 0 | 0 | |
| 1 | 0 | 1 | |
| 0 | 0 | 1 | Least Squares |
| 0 | 1 | 0 | |
| 0 | 1 | 1 | |
| 1 | 1 | 0 | |
| 1 | 1 | 1 | |