# Peer review of "Evolution of the ESA CCI Soil Moisture Climate Data Records and their underlying merging methodology"

_Earth System Science Data, 2019_

## Referee Comment (RC1) · Anonymous Referee #1 · 4 Mar 2019

**OVERVIEW**

The manuscript describes the different approaches used for merging multiple satellite soil moisture products in the ESA CCI project. Specifically, the three major versions of the ESA CCI soil moisture product are described, the methods used for merging active and passive product, and to finally obtain the combined products. The validation of the three versions with in situ observations is carried out, and the future steps for further

improving the product are outlined.

**GENERAL COMMENTS**
The paper is well written and clear. Different approaches have been developed for merging multiple satellite soil moisture products, using different sensors, and integration techniques. The ESA CCI soil moisture project is surely a significant initiative for building a long-term (currently 40 years) global scale soil moisture product, and the documentation of the different steps developed in the project in the scientific literature is needed, definitely. Therefore, I believe the paper deserves to be published on Earth System Science Data and I have some minor comments to be addressed.

1) The major differences in the ESA CCI soil moisture product versions are the different approaches for weighting the different single products in the merging (apart the addition of SMOS in v3). Therefore, in my opinion, the weight of the different sensors in the merging is the most important factor to be shown. For instance, in version 4 it seems that the major contribution is given by AMSR2 (from Figure 7), that is quite unexpected to me. Can the authors add some figures or statistics for showing the values of the weights in the different versions?

2) The results of the validation with in situ observations are briefly reported and it is not clear how large are the differences between the different product versions. For instance, plotting the difference in correlation with respect to the first version instead of the absolute values will put more evidence on the differences. Additionally, I suggest writing some numbers in the text, or in a Table, for showing the median/mean correlations for different periods and product versions.

**SPECIFIC COMMENTS**
Page 1, line 4: Add "iii)" before "a combined active-passive . . .".

Page 1, line 10: I suggest adding the expected date/month in which version 5 will be available.

Page 6, line 2: Change "as required" with "if required".

Page 6, line 30: "unweighted average". Is that correct? Likely better "equal weighted".

Page 9, line 32-33: It's not clear to me how the threshold is applied. What is the cumulative weight? I suggest better clarifying this part.

Page 15, lines 10-16: It is copy-paste with the abstract, I suggest changing.

Caption Tables 2 and 3: Change "p-vale" with "p-value".

**RECOMMENDATION**
On this basis, I found the paper relevant and useful and I suggest a minor revision before the publication in Earth System Science Data.

---

## Referee Comment (RC2) · Amen Al-Yaari (Referee) · 17 Mar 2019

OVERVIEW:

The paper by Gruber et al., presented the three major versions 2-4 of the ESA CCI soil moisture product and the different approaches used for the fusion of the satellite passive and active products. Finally, the three ESA CCI passive, active, and combined products were evaluated and compared to the ISMN in situ soil moisture ground-based measurements.

GENERAL COMMENTS:

It was a pleasure to read the well and clearly written manuscript. I only have few concerns that may make this manuscript more informative and enlightening:

-It would be interesting to see if there is also a difference in the representation of each product going from V2 to V4. In other words, the authors can add three maps for V2, V3, and V4 to show the spatial contribution of each satellite product.

- The CCI soil moisture team focused too much on improving the merging methodology but other aspects should be also considered such as forests regions which are masked in the current CCI versions, which limits the use of the ESA CCI SM product,and finding other methods that can avoid the use of the GLDAS-Noah land surface model.

- Also, I am surprised to read "This may be particularly problematic for trend analysis because such rescaling imposes any natural or spurious trends existing in the model to a certain degree on to the harmonised ESA CCI SM product" as it was always claimed that the trend of CCI is not affected by the GLDAS-Noah land surface model. In this regard, can the authors suggest for the end-users what kind of climate analysis that they do not recommend using the ESA CCI SM product (i. e. what are the cases where CCI will just reflect the GLDAS-Noah land surface model)?

- Finally, although the paper is focusing on the evolution of the CCI product, I would suggest the authors to add a "recommendations" section about the use of CCI product in applications, validations, .etc.

SPECIFIC COMMENTS:

In Figure 9 for some periods, V3 was giving slightly better correlations than V4 for the passive products. Can the authors comment on this?

RECOMMENDATION:

I found the document to be well written and well structured. The content provides valuable information about the ESA CCI product which is the achievement of significant team work. This paper is, therefore, worth publishing in the journal Earth System

[Figure]

Science Data (ESSD).

---

## Referee Comment (RC3) · Anonymous Referee #3 · 22 Mar 2019

The manuscript by Gruber et al. 2019 (ESSD) presents a comprehensive summary of the series of ESA CCI soil moisture data records. The ESA CCI SM is a successful initiative for providing long-term satellite-based soil moisture product for climate-related applications. The current manuscript is well written and already in a good shape. In general, I recommend it for publication after minor revision. My comments are summarized below and hope the authors could take them into account.

1. In Conclusion section, the high resolution SAR data is mentioned. I agree high-resolution soil moisture is in high demand, in particular, under the context that the spatial resolution of Earth system model is getting finer and finer. The Sentinel-1 data

[Figure]

can definitely provide data source for ESA CCI SM project. Upscaling and merging S-1 into ESA CCI SM is a promising direction, while downscaling is more questionable. Will the CCI SM team work towards improving soil moisture spatial resolution and release related CCI SM (high resolution) dataset in next version?

2. Another 'urgent' issue is the soil moisture under dense vegetated areas. In the manuscript, the mask applied for frozen areas are clearly stated. However, as far as I know, the highly vegetated areas are also masked out in the CCI SM. It might be better to discuss this issue and recommend future direction in the conclusion part.

3. Regarding the applications of CCI SM, the manuscript stated that the CCI SM is problematic for trend analysis. It might be misleading. Many studies have conducted trend analysis based on CCI SM. Rescaling against GLDAS-Noah indeed imposes the characteristic of model. However, the combined use of model and satellite data is not a problem, once you can guarantee it is a reliable product. It is better to revise those statements. From the user perspective, it is important to be clear the applicability of the product.

---

## Author Comment (AC1) · 7 May 2019

We thank Amen Al-Yaari and the two anonymous referees for their time and effort to review our manuscript, and for their very positive and constructive feedback which helped to further increase the quality of the paper. Below, all comments are addressed carefully.

Reviewer comments are marked in red.
Responses to the comments are marked in blue.
Cited changes that have been made in the manuscript are marked in *italic*.

[Figure]

Cited references are provided at the end of this response letter.

**Reviewer 1:**

The major differences in the ESA CCI soil moisture product versions are the different approaches for weighting the different single products in the merging (apart the addition of SMOS in v3). Therefore, in my opinion, the weight of the different sensors in the merging is the most important factor to be shown. For instance, in version 4 it seems that the major contribution is given by AMSR2 (from Figure 7), that is quite unexpected to me. Can the authors add some figures or statistics for showing the values of the weights in the different versions?

For versions 2 and 3 we show merging weights in Figure 4. Unfortunately, in version 4 they are not as straight forward to show because they dynamically change each day depending on which sensors provide valid observations at a particular location. This is the reason why we show the SNRs instead (Figure 7) from which the weights are derived. Given this figure, we do not quite understand why the reviewer got the impression that AMRS2 provides the largest contribution; SNRs of AMSR are much worse than of SMOS and ASCAT in large regions of Asia and Northern Europe, for example. Nevertheless, we agree that an indication of the relative contribution of each sensor would be very helpful. Therefore (also following the somewhat similar suggestion of the second reviewer) we have now included the new Figures 10 and 11 and a thorough discussion of this issue at the end of the evaluation section showing the relative contribution of individual sensors to valid measurements in the merged products, as well as overall data coverage.

The results of the validation with in situ observations are briefly reported and it is not clear how large are the differences between the different product versions. For instance, plotting the difference in correlation with respect to the first version

instead of the absolute values will put more evidence on the differences. Additionally, I suggest writing some numbers in the text, or in a Table, for showing the median/mean correlations for different periods and product versions.

We agree that the validation is relatively brief, but this is intentional as it is not a validation paper but a data set / merging methodology review. Therefore, our goal is to provide a general overview of quasi-global per-version-per-period product performance. A comprehensive review / compilation of ESA CCI SM validation studies is provided by *Dorigo et al.* (2015, 2017) as noted at the end of the evaluation section.

Regarding correlation differences, their interpretation is not meaningful without relating them to the absolute values because correlations are ratios and therefore affected by data set uncertainties in a non-linear fashion. For example, a correlation improvement of 0.1 from 0.5 to 0.6 corresponds to a noise reduction (which is the goal of the Least Squares merging) of about 40 % while the same improvement from 0.8 to 0.9 refers to a noise reduction of almost 60 % (see *Gruber et al.* (2016) for the relation between correlation coefficients and random errors). We therefore prefer to keep the absolute correlation plot as it indicates both absolute correlation values and their relative differences. We agree, however, that actual quantitative values are hard to infer from the figure. Therefore, following the Reviewer's suggestion, we have added a Table that summarizes the median correlation values right after Figure 9.

Page 1, line 4: Add "iii)" before "a combined active-passive : : :".

Done.

Page 1, line 10: I suggest adding the expected date/month in which version 5 will be available.

We added an expected rough time frame at the end of the introduction (where dates for all the other product versions are provided):

*"Moreover, an outlook to the expected developments that are planned for the next iteration, version 5 which is foreseen to be released in 2019, is provided."*

Unfortunately, we cannot be more specific than that, because due to technical issues the release has been postponed already, and it is not yet certain in which month it actually will be.

Page 6, line 2: Change "as required" with "if required".

Done.

Page 6, line 30: "unweighted average". Is that correct? Likely better "equal weighted".

Yes, "unweighted" is, according to two consulted native speakers, correct.

Page 9, line 32-33: It's not clear to me how the threshold is applied. What is the cumulative weight? I suggest better clarifying this part.

We have added an example to that paragraph to make the concept of the cumulative-weight-threshold more clear:

*"For example, assume that for merging AMSR-E, WindSat and SMOS into the PASSIVE product, their weights as derived from their relative SNR at a particular location are 0.1, 0.05 and 0.85, respectively. Because $N = 3$, the minimum-cumulative-weight threshold is 0.17. Therefore, if, at a particular day, only AMSR-E*

*and WindSat observations are available, which have a cumulative weight of 0.15, no soil moisture estimate is provided."*

Page 15, lines 10-16: It is copy-paste with the abstract, I suggest changing.

The paragraph has been shortened to be more concise and different from the abstract.

Caption Tables 2 and 3: Change "p-vale" with "p-value".

Done.

_**Reviewer 2 (Amen Al-Yaari):**_

It would be interesting to see if there is also a difference in the representation of each product going from V2 to V4. In other words, the authors can add three maps for V2, V3, and V4 to show the spatial contribution of each satellite product.

This is a good suggestion and quite similar to the first comment of Reviewer #1. We have added the new Figures 10 and 11, and a related discussion at the end of the evaluation section.

The CCI soil moisture team focused too much on improving the merging methodology but other aspects should be also considered such as forests regions which are masked in the current CCI versions, which limits the use of the ESA CCI SM product,and finding other methods that can avoid the use of the GLDAS-Noah land surface model.

We totally agree that these are two very important issues. Regarding forests, the ESA CCI SM team unfortunately has a very limited handle on that because it is more an issue of satellite soil moisture retrieval rather than product merging. In the latest version 4, we have introduced the SNR-VOD regressions, which obtains uncertainty estimates to properly merge all available soil moisture retrievals also over forests, instead of masking a product when it is suspected to be "unreliable". The only vegetation-masking that is actually part of the ESA CCI SM merging algorithm is that over tropical rainforests, which is arguably justified for microwave satellite measurements. We clarified this now in the beginning of Section 3:

_"Note that tropical rainforest areas are masked out in all ESA CCI SM products because microwave satellite measurements do not contain any useful soil moisture signal in these regions due to signal scattering and attenuation of the vegetation (Ulaby et al., 2014)."_
These regions are now also masked out in all provided figures. All other vegetation treatment takes place in the L2 soil moisture retrieval algorithms, which the ESA CCI SM team has no influence on. Note that we forgot to mention the VOD-based masking in the LPRM algorithm, we apologize for that. A sentence has been added to Section 3.2:

*"Soil moisture retrievals of all sensors are masked out if the VOD exceeds a certain threshold that depends on the microwave frequency of the respective sensor. For a complete description of LPRM and how it is applied see [...]"*

Although the ESA CCI SM team does not work on retrieval algorithms, both Vandersat and TU Wien are indeed continuously working on improving vegetation-related issues. Also, in Section 9, we point out that in the future we will hopefully be able to reassess the potential of other retrieval algorithms (e.g. official SMAP and SMOS products), which may have a better vegetation treatment over forests.

Regarding the avoidance of GLDAS-Noah, the last point of our "open-issues" list in the Conclusions section is indeed that it should be avoided and that we want to do that. Unfortunately, we cannot be more specific as to when it will happen, because the want-to and the can-do points in the algorithm development are a bit diverging depending on project budget.

Also, I am surprised to read "This may be particularly problematic for trend analysis because such rescaling imposes any natural or spurious trends existing in the model to a certain degree on to the harmonised ESA CCI SM product" as it was always claimed that the trend of CCI is not affected by the GLDAS-Noah land surface model. In this regard, can the authors suggest for the end-users what kind of climate analysis that they do not recommend using the ESA CCI SM product (i.

e. what are the cases where CCI will just reflect the GLDAS-Noah land surface model)?

We apologize that the formulation of this section was inaccurate. In this paper, we do not make statements conflicting earlier ESA CCI SM publications or claims about the properties of the result. It is correct that it was claimed that trends are not affected, but that statement always referred to the direction of trends and not to the absolute magnitude. We revised the paragraph accordingly:

*"This may impact long-term trend analyses because even though CDF-matching generally preserves the direction of an existing trend in a rescaled product, it can change its magnitude (Liu et al., 2012). That is, the rescaling against GLDAS-Noah can cause trends found in the harmonized ESA CCI SM product to appear stronger or weaker than they actually are. Moreover, the non-linear nature of the CDF-matching may introduce spurious error correlations, which could be problematic for TCA (see above) but also when evaluating the ESA CCI SM data set against other land surface models such as ERA-Interim/Land or MERRA2, which hampers a comprehensive large-scale validation of the product. A potential alternative could be the use of TCA-based linear rescaling, which was found to be potentially superior to CDF-matching especially for data merging if SNRs of different products are not equal (Yilmaz and Crow, 2013)."*

Finally, although the paper is focusing on the evolution of the CCI product, I would suggest the authors to add a "recommendations" section about the use of CCI product in applications, validations, .etc.

Such recommendations are exactly the purpose of *Dorigo et al.* (2015, 2017). We refer to them in the beginning of the paper (last paragraph of Section 1): "[...] product improvements have been validated in numerous publications and the data set has been proven to be useful in a large number of applications (for a comprehensive

review of these studies see *Dorigo et al.* (2015, 2017)) [...]". Since the paper is already quite lengthy, we would prefer to avoid this redundancy.

In Figure 9 for some periods, V3 was giving slightly better correlations than V4 for the passive products. Can the authors comment on this?

Since correlation percentiles of the PASSIVE product are consistently non-significantly different or higher in v4 we believe the reviewer was referring to the noticeably degraded lower quartile/whisker of the COMBINED product in the third merging period? We have added a paragraph on that to the discussion of Figure 9:

*"The lower quartile of anomaly correlations of the COMBINED product in the same merging period has slightly degraded from ESA CCI SM v3 to v4. This may be caused by an inaccurate VOD-based weight prediction in the v4 product as this merging period does not cover most of summer and autumn retrievals while weight prediction is based on annual-average VOD conditions. However, it may also just be a statistical artefact given the significantly reduced data coverage in this period and the reduced number of stations available to calculate correlation percentiles."*

*Reviewer 3:*

In Conclusion section, the high resolution SAR data is mentioned. I agree high-resolution soil moisture is in high demand, in particular, under the context that the spatial resolution of Earth system model is getting finer and finer. The Sentinel-1 data can definitely provide data source for ESA CCI SM project. Upscaling and merging S-1 into ESA CCI SM is a promising direction, while downscaling is more questionable. Will the CCI SM team work towards improving soil moisture spatial resolution and release related CCI SM (high resolution) dataset in next version?

Yes, the ESA CCI team will work towards including SAR data in the future, but this will not happen in the next product version 5. Unfortunately, we cannot reliably predict yet when we will be able to do so as it largely depends on the future funding situation.

Another 'urgent' issue is the soil moisture under dense vegetated areas. In the manuscript, the mask applied for frozen areas are clearly stated. However, as far as I know, the highly vegetated areas are also masked out in the CCI SM. It might be better to discuss this issue and recommend future direction in the conclusion part.

[Reviewer #2 has made a very similar comment. The following response is therefore largely identical to our reply there]: The ESA CCI SM team unfortunately has a very limited handle on that because it is more an issue of satellite soil moisture retrieval rather than product merging. In the latest version 4, we have introduced the SNR-VOD regressions, which obtains uncertainty estimates to properly merge all available soil moisture retrievals also over forests, instead of masking a product when it is suspected to be "unreliable". As the reviewer correctly points out, the only vegetation-masking that is applied now in the ESA CCI SM algorithm is that over tropical rainforests, which is arguably justified for microwave satellite measurements. We clarified this now in the

beginning of Section 3:

*"Note that tropical rainforest areas are masked out in all ESA CCI SM products because microwave satellite measurements do not contain any useful soil moisture signal in these regions due to signal scattering and attenuation of the vegetation (Ulaby et al., 2014)."*

These regions are now also masked out in all provided figures. All other vegetation treatment takes place in the L2 soil moisture retrieval algorithms, which the ESA CCI SM team has no influence on. Note that we forgot to mention the VOD-based masking in the LPRM algorithm, we apologize for that. A sentence has been added to Section 3.2:

*"Soil moisture retrievals of all sensors are masked out if the VOD exceeds a certain threshold that depends on the microwave frequency of the respective sensor. For a complete description of LPRM and how it is applied see [...]"*

Although the ESA CCI SM team does not work on retrieval algorithms, both Vandersat and TU Wien are indeed continuously working on improving vegetation-related issues. Also, in Section 9, we point out that in the future we will hopefully be able to reassess the potential of other retrieval algorithms (e.g. official SMAP and SMOS products), which may have a better vegetation treatment over forests.

Regarding the applications of CCI SM, the manuscript stated that the CCI SM is problematic for trend analysis. It might be misleading. Many studies have conducted trend analysis based on CCI SM. Rescaling against GLDAS-Noah indeed imposes the characteristic of model. However, the combined use of model and satellite data is not a problem, once you can guarantee it is a reliable product. It is better to revise those statements. From the user perspective, it is important to be clear the applicability of

the product.

[The following response is identical to the one we provided to Reviewer #2, who made the same comment]: We apologize that the formulation of this section was inaccurate. In this paper, we do not make statements conflicting earlier ESA CCI SM publications or claims about the properties of the result. It is correct that it was claimed that trends are not affected, but that statement always referred to the direction of trends and not to the absolute magnitude. We revised the paragraph accordingly:

*"This may impact long-term trend analyses because even though CDF-matching generally preserves the direction of an existing trend in a rescaled product, it can change its magnitude (Liu et al., 2012). That is, the rescaling against GLDAS-Noah can cause trends found in the harmonized ESA CCI SM product to appear stronger or weaker than they actually are. Moreover, the non-linear nature of the CDF-matching may introduce spurious error correlations, which could be problematic for TCA (see above) but also when evaluating the ESA CCI SM data set against other land surface models such as ERA-Interim/Land or MERRA2, which hampers a comprehensive large-scale validation of the product. A potential alternative could be the use of TCA-based linear rescaling, which was found to be potentially superior to CDF-matching especially for data merging if SNRs of different products are not equal (Yilmaz and Crow, 2013)."*

**References**

Dorigo, W., A. Gruber, R. De Jeu, W. Wagner, T. Stacke, A. Loew, C. Albergel, L. Brocca, D. Chung, R. Parinussa, et al. (2015), Evaluation of the ESA CCI soil moisture product using ground-based observations, *Remote Sensing of Environment*, **162**, p. 380–395.

Dorigo, W., W. Wagner, C. Albergel, F. Albrecht, G. Balsamo, L. Brocca, D. Chung, M. Ertl, M. Forkel, A. Gruber, et al. (2017), ESA CCI soil moisture for improved earth system understanding: state-of-the art and future directions, *Remote Sensing of Environment*, **203**, p. 185–215, doi:10.1016/j.rse.2017.07.001.

Gruber, A., C.-H. Su, S. Zwieback, W. Crow, W. Dorigo, and W. Wagner (2016), Recent advances in (soil moisture) triple collocation analysis, *International Journal of Applied Earth Observation and Geoinformation*, **45**, p. 200–211, doi:10.1016/j.jag.2015.09.002.

Liu, Y., W. Dorigo, R. Parinussa, R. de Jeu, W. Wagner, M. McCabe, J. Evans, and A. van Dijk (2012), Trend-preserving blending of passive and active microwave soil moisture retrievals, *Remote Sensing of Environment*, **123**(0), p. 280–297, doi:10.1016/j.rse.2012.03.014.

Ulaby, F. T., D. G. Long, W. J. Blackwell, C. Elachi, A. K. Fung, C. Ruf, K. Sarabandi, H. A. Zebker, and J. Van Zyl (2014), *Microwave radar and radiometric remote sensing*, vol. 4, University of Michigan Press Ann Arbor.

Yilmaz, M. T., and W. T. Crow (2013), The optimality of potential rescaling approaches in land data assimilation., *Journal of Hydrometeorology*, **14**(2), doi:10.1175/JHM-D-12-052.1.

[Figure]

PASSIVE COMBINED

v02.2

v02.2

v03.3

v03.3

v04.4

v04.4

0.0    0.2    0.4    0.6    0.8    1.0

**Fig. 1.** (New Figure 10): Fraction of days during the latest four merging periods where the PASSIVE and COMBINED products version v02.2, v03.3 and v04.4 provide valid observations.

[Figure]

**Fig. 2.** (New Figure 11): Fractional contribution of the individual sensors to valid ESA CCI SM soil moisture estimates of the PASSIVE and COMBINED products during the last four merging periods.